# Convergent trends and spatiotemporal patterns of Aedes-borne arboviruses in Mexico and Central America

**Bernardo Gutierrez[1,2]\***, **Darlan da Silva Candido[1,3]**, **Sumali Bajaj[1]**, **Abril Paulina Rodriguez Maldonado[4]**, **Fabiola Garces Ayala[4]**, **María de la Luz Torre Rodriguez[4]**, **Adnan Araiza Rodriguez[4]**, **Claudia Wong Arámbula[4]**, **Ernesto Ramírez González[4]**, **Irma López Martínez[4]**, **José Alberto Díaz-Quiñónez[4,5]**, **Mauricio Vázquez Pichardo[4]**, **Sarah C. Hill[6]**, **Julien Thézé[7]**, **Nuno R. Faria[1,3,8,9]**, **Oliver G. Pybus[1,6]**, **Lorena Preciado-Llanes[10]**, **Arturo Reyes-Sandoval[10,11]**, **Moritz U. G. Kraemer[1]\***, **Marina Escalera-Zamudio[1]\***

**1** Department of Biology, University of Oxford, Oxford, United Kingdom, **2** Colegio de Ciencias Biológicas y Ambientales, Universidad San Francisco de Quito USFQ, Quito, Ecuador, **3** Instituto de Medicina Tropical, Faculdade de Medicina da Universidade de São Paulo, São Paulo, Brazil, **4** Instituto de Diagnóstico y Referencia Epidemiológicos (InDRE) "Dr. Manuel Martínez Báez", Secretaría de Salud, Mexico City, México, **5** Instituto de Ciencias de la Salud, Universidad Autónoma del Estado de Hidalgo, Pachuca de Soto, Mexico, **6** Department of Pathobiology and Population Sciences, Royal Veterinary College, London, United Kingdom, **7** Université Clermont Auvergne, INRAE, VetAgro Sup, UMR EPIA, Saint-Genès-Champanelle, France, **8** MRC Centre for Global Infectious Disease Analysis, School of Public Health, Imperial College London, London, United Kingdom, **9** The Abdul Latif Jameel Institute for Disease and Emergency Analytics, School of Public Health, Imperial College London, London, United Kingdom, **10** Nuffield Department of Medicine/ Wellcome Centre for Human Genetics, University of Oxford, Oxford, United Kingdom, **11** Instituto Politécnico Nacional (IPN), Av. Luis Enrique Erro s/n., Unidad Adolfo López Mateos, Mexico City, Mexico

\* bernardo.gutierrez@biology.ox.ac.uk (BG); moritz.kraemer@biology.ox.ac.uk (MUGK); marina. escalerazamudio@biology.ox.ac.uk (MEZ)

**Data Availability Statement:** Virus sequences generated in this study are provided as alignments in S1 Data (CHIKV), 2 (DENV-1) and 3 (DENV-2).

## Abstract

### Background

Aedes-borne arboviruses cause both seasonal epidemics and emerging outbreaks with a significant impact on global health. These viruses share mosquito vector species, often infecting the same host population within overlapping geographic regions. Thus, comparative analyses of the virus evolutionary and epidemiological dynamics across spatial and temporal scales could reveal convergent trends.

### Methodology/Principal findings

Focusing on Mexico as a case study, we generated novel chikungunya and dengue (CHIKV, DENV-1 and DENV-2) virus genomes from an epidemiological surveillance-derived historical sample collection, and analysed them together with longitudinally-collected genome and epidemiological data from the Americas. Aedes-borne arboviruses endemically circulating within the country were found to be introduced multiple times from lineages predominantly sampled from the Caribbean and Central America. For CHIKV, at least thirteen introductions were inferred over a year, with six of these leading to persistent

GenBank Accession numbers for the publicly available sequences used in this study are listed in the S4 Data.

**Funding:** This work was supported by the United Kingdom Research & Innovation office and the Department of Health and Social Care using UK Aid funding (ARS: BSRC/EPSRC/NIHR 971557); John Fell OUP Research Fund Award (MEZ and MUGK: grant 0008724); Wellcome Trust Sir Henry Wellcome Postdoctoral Fellowship (SCH: 220414/Z/20/Z, received salary from this funding); the European Union Horizon 2020 project MOOD (MUGK and JT: #874850); Leverhulme Trust ECR Fellowship (MEZ: ECF-2019-542, received salary from this funding). NRF acknowledges support from Bill and Melinda Gates Foundation (NRF: INV-034540) and Medical Research Council-Sao Paulo Research Foundation (FAPESP) CADDE partnership award (NRF: MR/S0195/1 and FAPESP 18/14389-0). OGP and MUGK acknowledge support from the Oxford Martin School. The views expressed in this publication are those of the author(s) and not necessarily those of the Department of Health and Social Care. The funders had no role in study design, data collection and analysis, decision to publish, or preparation of the manuscript.

**Competing interests:** The authors have declared that no competing interests exist.

transmission chains. For both DENV-1 and DENV-2, at least seven introductions were inferred over a decade.

## Conclusions/Significance

Our results suggest that CHIKV, DENV-1 and DENV-2 in Mexico share evolutionary and epidemiological trajectories. The southwest region of the country was determined to be the most likely location for viral introductions from abroad, with a subsequent spread into the Pacific coast towards the north of Mexico. Virus diffusion patterns observed across the country are likely driven by multiple factors, including mobility linked to human migration from Central towards North America. Considering Mexico's geographic positioning displaying a high human mobility across borders, our results prompt the need to better understand the role of anthropogenic factors in the transmission dynamics of Aedes-borne arboviruses, particularly linked to land-based human migration.

### Author summary

Mexico is endemic to several Aedes-borne arboviruses relevant to global health, and ranks within the top five countries in the Americas that report the highest case numbers. Our study provides a general overview of arbovirus introduction, spread and establishment patterns in North and Central America, and should be of interest to both local health and global authorities. Moreover, it sets to explore the paradigm of convergence at different scales in independent virus populations, represented by comparable epidemiological and evolutionary trends in Aedes-borne arboviruses sharing ecological niches. Our results represent important advances in the study of mosquito-borne viruses listed as a threat to global health, specifically applied to key countries within the developing world.

## Introduction

Aedes-borne arboviruses spread is driven by a complex interaction between environmental conditions [1], ecological factors affecting vector populations [2], human behaviour and mobility [3–8], as well as pre-existing levels of immunity within the host population [9–11]. Whilst some viruses display seasonal dynamics with varying transmission peaks across time [12,13], novel and/or re-emerging viruses can cause explosive outbreaks in areas where the local population has limited or non-existent prior immunity [14–17]. Thus, untangling the individual contributions of the diverse drivers impacting viral spread across spatiotemporal scales remains a challenge. However, comparative phylodynamic approaches can provide a powerful tool to identify shared epidemiological and evolutionary trends, offering valuable information to better understand current and future outbreaks. Moreover, mapping arboviral emergence and spread within specific regions can inform on the development of efficient, spatially targeted interventions, including widening genomic-epidemiology surveillance efforts, local vector control/clearance, and vaccination campaigns.

Apart from sharing vector species, Aedes-borne arboviruses tend to circulate within the same host population and geographic region, and thus are expected to exhibit similar evolutionary and epidemiological trends (despite their specific disease dynamics likely to be shaped by immunity within the host population [18–20]). Highlighted by the more recent

introduction of the Zika (ZIKV) and chikungunya (CHIKV) mosquito-borne viruses into the Americas (compared to DENV), contrasting the evolutionary and epidemiological dynamics of emerging and established viruses that have circulated endemically over multiple decades within the same region, could help identify recurrent patterns in Aedes-borne arboviruses sharing ecological niches.

In the Americas (ruling out possible unidentified chikungunya fever cases occurring during the 17[th] and 19[th] centuries, likely to be misdiagnosed as dengue fever [21,22]), locally acquired CHIKV infections were first confirmed during the mid 2010s [23], after which large outbreaks were detected between 2013 to 2017. On the other hand, all DENV serotypes (DENV-1 to DENV-4) display a somewhat different epidemic history within the region. Dengue-like illness has been well documented since the late 1700s [24], yet a considerable epidemiological shift occurred after the establishment of larger and more frequent outbreaks recorded during the 1950-60s [25]. By the 1980-90s, DENV outbreaks were reported on a yearly basis in at least 24 countries, with virus re-emergence mostly driven by the re-establishment of the previously eradicated *Aedes aegypti* mosquito population following the ban of use of DDT in the 1970s [25] (for more details, see **S1 Text**).

In Mexico, arbovirus surveillance takes place under the National Epidemiological Surveillance System ('Sistema Nacional de Vigilancia Epidemiológica–SINAVE') [26]. SINAVE involves nationwide case reporting by hospitals and clinics within and outside Mexico's Ministry of Health's healthcare network. Employing a sentinel approach based on positive case definition, healthcare institutions select and ship samples (along with essential information such as patient demographics, symptoms, and initial laboratory test results) to designated State Public Health Laboratories (LESP) for diagnosis confirmation using reference cellular and molecular biology standards. Subsequently, positive samples are sent from the LESP network to InDRE (Instituto de Diagnóstico y Referencia Epidemiológicos Dr. Manuel Martínez Báez, Ministry of Health of Mexico, Mexico City) for long-term storage. This enables a centralized historical collection spanning multiple decades and locations, providing a longitudinal perspective for the surveillance of infectious diseases in Mexico. The General Directorate of Epidemiology (Dirección General de Epidemiología) oversees SINAVE in centralizing and compiling data on confirmed cases reported nationwide, in order to facilitate the development of outbreak response measures and the dissemination of epidemiological information. Specifically, for Aedes-borne virus surveillance, blood and serum samples are typically shipped to the National Arbovirus Reference Laboratory, a PAHO-WHO collaborating centre at InDRE [26].

CHIKV surveillance formally commenced in late 2014, with a high number of cases detected during 2015, particularly in the southern and southwestern states of the country [27]. On the other hand, the longitudinal surveillance of DENV shows that serotypes 1, 2 and 4 co-circulated in the country since the 1980s [28], whilst DENV-3 was first detected in 1995 [29]. Given ideal ecological and climatic conditions favouring vector populations, the south coast region of the country (comprising the states of Chiapas, Oaxaca, Guerrero, Veracruz, Tamaulipas, Quintana Roo, Campeche and Yucatán) has been historically most affected [28,29]. Of interest, the spread of CHIKV and ZIKV across the country coincided with a decrease in DENV incidence observed for these states, with a low in cases recorded during the peak of both the CHIKV and ZIKV epidemics [30].

Historical samples linked to cases officially reported under SINAVE represents a valuable tool for retrospectively exploring epidemiological and evolutionary dynamics of Aedes-borne arboviruses circulating in Mexico. In this light, we present a new set of 39 CHIKV, 7 DENV-1 and 11 DENV-2 partial or complete viral genomes derived from samples collected between 2013 and 2017 across different states of the country. Under the hypothesis that the evolutionary and epidemiological dynamics of different Aedes-borne arboviruses circulating in the

Americas may display converging trends, we used a phylogeographic approach to analyse diversely sourced genome data from the region (including the genomes generated here), with a particular focus on quantifying lineage importations into Mexico. By interpreting our results alongside longitudinal epidemiological data, we compare the spatial epidemiology of these three viruses, and further assess extrinsic factors that are likely driving their dynamics. We find important similarities between the introduction of DENV and CHIKV into the country across different spatial and temporal scales.

## Methods

### Ethics statement

We declare that under the authority of the ethics committee "Comité de ética e investigación del ICSa–UAEH" (Instituto de Ciencias de la Salud/Universidad Autónoma del Estado de Hidalgo, Mexico), our study has been granted ethical approval registered under the following number: ICSa 157/2023 (signed by the president Dra. Itzia Maria Cazares Palacios on March 6th 2023). Patient consent was not obtained/required, given that the sample and data utilized in this study was already anonymized. The study involved no direct interaction with patients.

**Sample selection and virus genome sequencing.** Derived from the historical Aedes-borne case sample collection available at InDRE, we sequenced a set of both human and mosquito-derived DENV-1, DENV-2 and CHIKV virus genomes from samples selected based on their geographic location and collection date (S1 Table). Similar to neighbouring countries, DENV-1 and DENV-2 have prevailed in Mexico since the early 2000s [31,32]. Consequently, historical samples for other DENV serotypes (specifically DENV-3 and DENV-4) are scarce, further limiting the possibility of performing in-depth genomic epidemiology-based analyses. Thus, DENV-3 and DENV-4 samples were excluded. Initially, ZIKV samples were also considered, but failed to amplify under the multiplexed PCR described below. Briefly, candidate samples were first selected by identifying those with collection dates and locations poorly represented by publicly available genomic data from Mexico (https://www.ncbi.nlm.nih.gov/genbank/). Samples were further selected based on RNA/cDNA availability and initial diagnostic qPCR $C_t$ values ($C_t \leq 35$). Total RNA was used to generate cDNA using the SuperScript™ IV First-Strand Synthesis System (Invitrogen, CA, USA) with random hexamers, and subjected to multiplexed PCR reactions using virus-specific primer sets [33,34]. PCR products consisted of multiple overlapping ~400 nt fragments that cover the vast majority of the viral coding regions (excluding extreme ends). PCR reactions were performed using the Q5 High-Fidelity DNA polymerase kit (New England Biolabs, MA, USA) for 35–40 cycles, whilst products were purified using AmpureXP magnetic bead system (Beckman Coulter, CA, USA) to be quantified on the Qubit 3.0 using the Qubit dsDNA High Sensitivity kit (Life Technologies, CA, USA).

For viral genome sequencing, we used a ligation sequencing approach [35] on the MinION sequencing device (Oxford Nanopore Technologies, UK). Sequencing libraries were prepared from the purified PCR products using barcoding for individual samples (EXP-NBD104, EXP-NBD114, and SQK-LSK109 kits, Oxford Nanopore Technologies, UK). For sequencing, samples were pooled equimolarly to be loaded onto R9.4 flow cells (Oxford Nanopore Technologies, UK). All runs were carried out until a sequencing depth of >20X was achieved, monitoring in real-time using the RAMPART platform (https://artic.network/ncov-2019/ncov2019-using-rampart.html). Individual consensus viral genomes were assembled against designated reference sequences for each serotype under a >95% sequence identity threshold, whilst their assignment to specific viral genotypes was performed using the Arbovirus Typing

Tools from the Genome Detective web server [36]. Virus genome sequences generated in this study are available as alignments provided in **S1–S3 Data.**

**Phylogenetic analyses.** To generate the datasets used for analyses, complete viral genome sequences from the Americas belonging to DENV-1 (Genotype V), DENV-2 (Genotype III) and CHIKV (Asian Genotype) available in GenBank (https://www.ncbi.nlm.nih.gov/genbank/) were retrieved. High-quality sequences were included if collected from any country in North, Central and South America or the Caribbean as of 2020-02-01, and if >10,000 nt long. The DENV dataset was further expanded by adding 16 DENV-1 and 14 DENV-2 complete genomes from Nicaragua collected in between 2013 and 2019 (derived from a paediatric cohort that only became publicly available after initial sequence collation) [37,38]. Our final datasets include a total of 420 CHIKV, 375 DENV-1, and 643 DENV-2 genome sequences, selected to represent all countries in North and Central America with virus genome representation sampled though time. Given the reduced size of our datasets, it was preferable to employ all available sequences rather than subsampling and risking loss of phylogenetic signal. However, quality control measures implemented included validating inferred phylogenetic trees by comparing general evolutionary trends with publicly accessible phylogenies, in order to detect inconsistencies in known divergence patterns both at broader and deeper node levels. Datasets encompassed both sequences generated in our study and those obtained from public databases (accession numbers available in **S4 Data**). The geographical distribution of the sequences sampled for all analyses can be found in **S1 Fig.**

Each dataset was aligned using *MAFFT* [39], visually inspected, and further trimmed to remove untranslated terminal genomic regions (UTRs). Final alignments comprised a total length of 11,151 bp for CHIKV, 10,180 bp for DENV-1 and 10,023 bp for DENV-2. Maximum likelihood (ML) phylogenetic trees were then constructed using *IQtree 2.0* [40], resulting in individual phylogenetic trees for each virus. ML trees were inferred under a General Time Reversible (GTR) substitution model, whilst the substitution rate heterogeneity across sites was modelled under a Gamma distribution. Branch support was estimated using non-parametric Shimoda-Hasegawa approximate Likelihood Ratio Tests (SH-aLRTs) and 1000 replicates [41]. The resulting trees were midpoint rooted and annotated with the sequence collection date and location (country/region). To assess the temporal signal within the trees, the correlation between tree tips to the root and associated collection dates was visualised as linear regressions using *TempEst v1.5.3* [42].

**Time-calibrated phylogeographic analyses.** To further investigate the spatial and temporal dynamics of the viruses studied here, time-calibrated phylogenies were estimated from the alignments described above with *BEAST v1.10.4* [43], using sequence collection dates to inform tree tips. For two DENV-1 genomes with no collection date available, a uniform prior was assigned to the corresponding tips. All trees were inferred using the Hasegawa-Kishino-Yano (HKY) substitution model with rate heterogeneity modelled under a Gamma distribution. A Skyride tree prior [44] and a relaxed uncorrelated clock model [45] under a continuous-time Markov chain (CTMC) rate prior were further implemented [46]. Markov Chain Monte Carlo (MCMC) chains were run in duplicate for $1 \times 10^8$ states, with the first $1 \times 10^7$ discarded as a burn-in. Independent runs were combined using *LogCombiner*, resulting in empirical tree distributions that were summarised into a Maximum Clade Credibility (MCC) tree using *TreeAnnotator*. Convergence for individual MCMC chains was assessed using *Tracer* [47], verifying that all relevant parameters reached an effective sample size (ESS) value of >200. In order to evaluate changes in viral effective population size (Ne) across time under a coalescent model [48], for DENV-2, Tracer was used to generate a Bayesian Skyline plot (BSP) derived from the MCC tree [46]

Virus diffusion patterns within Mexico and across neighbouring countries was explored using a phylogeographic approach under a discrete trait analysis (DTA) [49]. Locations assigned to tips correspond either to the country of collection, or to the geographic region within the country where the sequences were collected from (for Mexico sequences only, following the eight-region model as a prevailing standard) (**S1 Fig**). For this, posterior tree distributions were resampled under an asymmetric substitution model for location reconstruction at ancestral nodes [49]. The statistical significance of transition rates between locations that best explain viral diffusion processes was evaluated under a Bayes Factor (BF) test, explored through a Bayesian Stochastic Search Variable Selection (BSSVS) implemented in *BEAST* [49]. A criterion of a BF > 4 was used to define well-supported diffusion rates, together with a corresponding posterior probability (PP) of >0.5 [50]. The expected number of transitions between countries and regions was estimated for each virus subtype using a robust counting approach [51]. As in the previous step, MCC trees were generated by summarizing the posterior tree sample, further estimating posterior probabilities for inferred locations at given nodes.

**Collation and analysis of historical epidemiological data.** The SINAVE system collects arbovirus data at an individual case level using a digital platform in real time, made available to us through the National Institute for Epidemiological Diagnosis and Reference (InDRE) [26]. This database includes over 200 variables for officially confirmed DENV and CHIKV cases recorded between 2010 and 2019, including: clinical (date of symptom onset, date of sample collection, final diagnosis), demographic (patient age and sex) and geographic information (collection state, municipality and locality). It also included serotype information for a proportion (approximately 10%) of positive DENV cases confirmed by serology or by PCR product sequencing. Derived from the database, different variables were used to explore the epidemiological trends across time of the viruses studied here.

## Results

### CHIKV genome data

From selected CHIKV samples collected from between May 2014 to December 2015 and 18/32 (56%) federal states in Mexico, we were able to generate 39 complete CHIKV genomes that were assigned to the Asian genotype, corresponding to the main genotype reported to circulate in the Americas [52] (**S1 Fig** and **S1 Table**). The set of newly generated viral genomes represent a substantial addition to previous publicly available data from Mexico, enhancing the geographic and temporal sampling range for CHIKV in the country (**Table 1**). In this context, throughout the CHIKV outbreak in the Americas, genome availability across countries gradually increased through 2014, but was maintained at <10 sequences per month (**S2 Fig**). Virus genomes from the USA, various Caribbean territories and Central American nations were made publicly available in that year, yet a limited number of genome sequences were generated from Mexico between 2014 and 2015. Overall, the number of CHIKV genomes per country does not correlate with the cumulative number of cases reported over time (**S3 Fig**), indicating a poor association between virus genome and epidemiological data.

### Epidemiological dynamics of CHIKV in Central America and Mexico

The first CHIKV cases were reported in the Eastern Caribbean (also named the Lesser Antilles (**S2 Table**), likely leading to larger outbreaks posteriorly observed in other Central Caribbean islands (**Fig 1A**) [53–60]. During the early phase of the initial outbreak, the Dominican Republic (Central Caribbean) and Guadeloupe (Eastern Caribbean) reported over 15,000 new daily cases at their respective epidemic peaks occurring around July and December 2014 (**S1A Fig**). Central America and the Andean region of South America also experienced large outbreaks in

**Table 1. Aedes-borne arbovirus epidemiology and genome sampling in Mexico.**

| Region | State | CHIKV | | DENV-1 | | DENV-2 | |
|---|---|---|---|---|---|---|---|
| | | Cases[a]* | Seqs[c] | Expected cases[b]** | Seqs[c] | Expected cases[b]** | Seqs[c] |
| Northwest | Baja California Norte | 905 | 2 | 16703 | – | 9522 | – |
| | Baja California Sur | | | | | | |
| | Chihuahua | | | | | | |
| | Durango | | | | | | |
| | Sinaloa | | | | | | |
| | Sonora | | | | | | |
| Northeast | Coahuila | 240 | 4 | 6618 | – | 14207 | – |
| | Nuevo León | | | | | | |
| | Tamaulipas | | | | | | |
| Centre-north | Aguascalientes | 31 | 3 | 6749 | – | 314 | – |
| | Guanajuato | | | | | | |
| | Querétaro | | | | | | |
| | San Luis Potosí | | | | | | |
| | Zacatecas | | | | | | |
| Centre-south | Mexico state | 795 | 6 | 5613 | 1 | 1150 | 1 |
| | Mexico City | | | | | | |
| | Morelos | | | | | | |
| West | Colima | 2959 | 7 | 20145 | 3 | 8839 | 4 |
| | Jalisco | | | | | | |
| | Michoacán | | | | | | |
| | Nayarit | | | | | | |
| East | Hidalgo | 2501 | 6 | 17267 | 1 | 10225 | 4 |
| | Puebla | | | | | | |
| | Tlaxcala | | | | | | |
| | Veracruz | | | | | | |
| Southwest | Chiapas | 3951 | 12 | 9994 | 1 | 16150 | 6 |
| | Guerrero | | | | | | |
| | Oaxaca | | | | | | |
| Southeast | Campeche | 2276 | 7 | 12839 | 70 | 6384 | 11 |
| | Quintana Roo | | | | | | |
| | Tabasco | | | | | | |
| | Yucatán | | | | | | |

*Source: SINAVE data, InDRE/Ministry of Health, Mexico.

**Calculated as the relative proportion of the serotype relative to all serotyped cases multiplied by the total of reported DF/DHF cases, rounded up. Source: SINAVE data, InDRE/Ministry of Health, Mexico.

[a]Reported between 2015 and 2018.

[b]Reported between 2013 to 2018.

[c]Including genomes generated in this study, plus previous publicly available genome data from Mexico.

2015 and early 2016 but were generally associated with lower case numbers (**Fig 1A**). Between March and December 2015, Colombia reported the largest number of cases within the region (**S2A Fig**). Subsequently, the southern cone region of South America (represented predominantly by Brazil) drove a resurgence of CHIKV cases in the continent over the following year (between March 2016 and January 2017) (**Figs 1A** and **S2A**) [34,53,56,61–64].

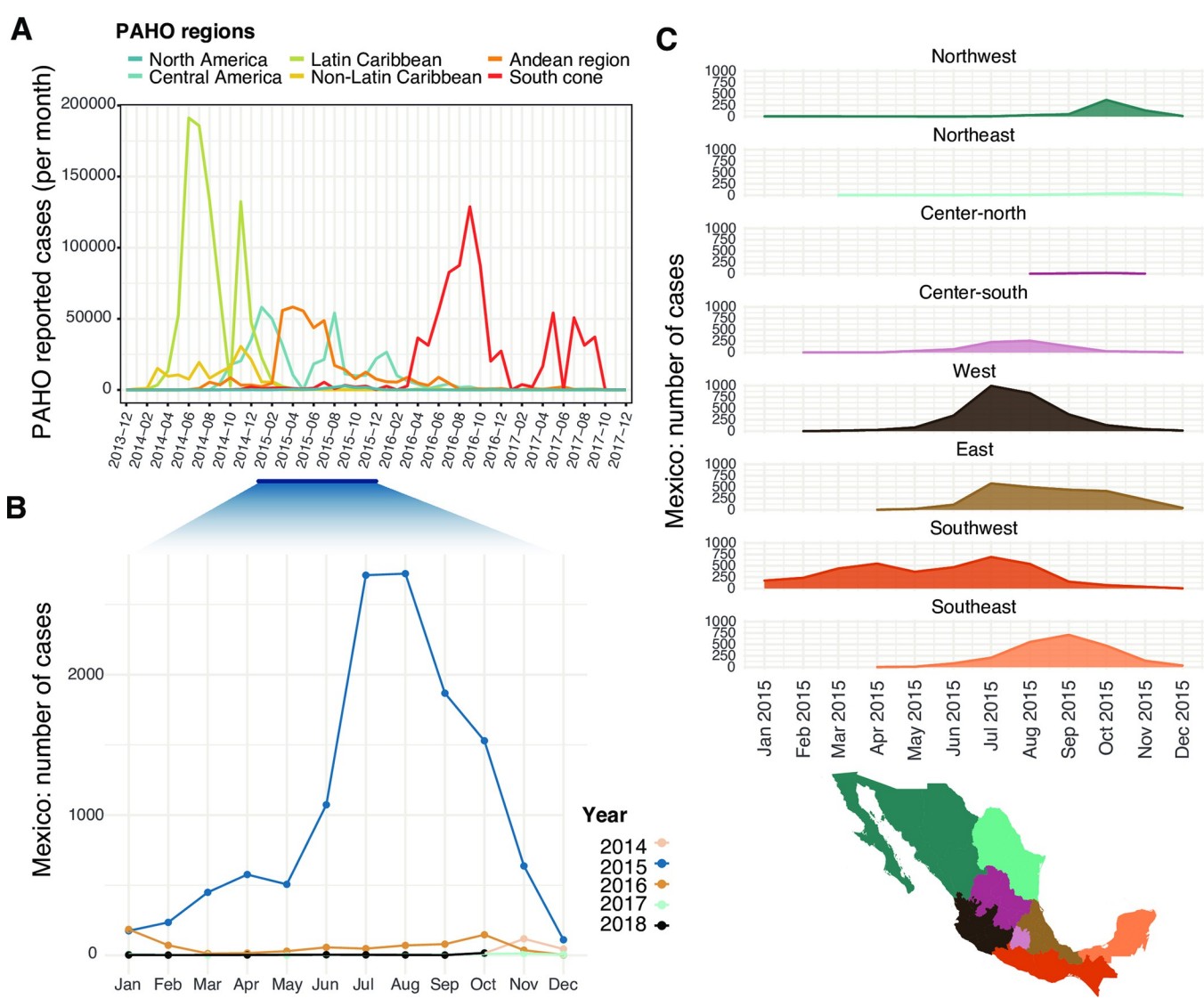

**Fig 1. Epidemiological trends of CHIKV in the Americas from 2013 to 2018.** (A) Monthly number of CHIKV cases reported to the Pan-American Health Organisation (PAHO) between 2013 and 2017, grouped by PAHO region. The largest epidemic peak in Mexico (PAHO North America region) occurred in 2015, highlighted in blue. (B) Monthly number of confirmed CHIKV cases in Mexico grouped by year, as reported by the SINAVE. (C) Monthly number of confirmed CHIKV cases per geographic region in Mexico during 2015. A map of the states included in each geographic region is shown below. Details of the states included in each region are available in Table 1. Plots were generated using the ggplot package (https://ggplot2.tidyverse.org/index.html) in R. Original base layer maps use as a source for geospatial base layer data public domain maps imported from the Natural Earth data project (https://cran.r-project.org/web/packages/rnaturalearth/index.html).

 The CHIKV epidemic in Mexico began approximately one year after the first outbreak recorded in the Caribbean islands in late 2013 [65,66]. Mexico reported CHIKV cases from end 2014 to 2018, with new cases rapidly increased after January 2015 to reach a highest during the summer of that year (**Fig 1B**), with numbers growing from 506 (recorded in May) to around 2700 (recorded in August 2015). CHIKV cases increased in the Southwest region of the country, followed by case peaks observed in the East (Gulf of Mexico) and West (Pacific Ocean) coast regions, as well as in the Southeast (encompassing the Yucatán peninsula) (**Fig 1C**). Limited numbers of cases were observed in the centre-south region (comprising Mexico City and surrounding areas). Later, a small peak in cases was observed in the

Northwest, representing the only region that saw a modest second wave of CHIKV cases during January 2017 (**S4 Fig**).

## Evolutionary dynamics of CHIKV in Central America and Mexico

Our phylogenetic analysis revealed that the CHIKV outbreak in the Americas was caused by a single virus lineage (SH-aLRT = 97.9, referred to here as the 'American' lineage). This lineage directly descends from sequences sampled from Southeast Asia and the Pacific Islands (**Fig 2A**), denoting a single introduction event of the CHIKV Asian genotype into the Americas. An early rapid expansion of CHIKV across North and South America is apparent at the base of the lineage, supported by a poor clustering pattern observed for sequences collected from different countries (**S5 Fig**). Within the 'American' lineage, a large clade containing genome sequences predominantly sampled from the Caribbean, Central America and North America was identified, including all sequences from Mexico (referred here to as the CCNA clade. SH-aLRT = 77.5). Within the CCNA, the CHIKV genomes from Mexico mostly group into 6 well-supported clusters, grouping together other sequences from Central America and the Caribbean (**Figs 2** and **S5**).

Consistent with previous observations, we estimated an earliest TMRCA date for the root of the CCNA clade going back to mid-September 2013 (Median = 2013.6941, 95% HPD = 2013.5867–2013.7865 (**Fig 2A**). The most basal branches within the CCNA clade show early viral circulation predominantly within the Eastern Caribbean, suggesting multiple exportations into the Central Caribbean, South, Central and North America region. This includes two importations into Mexico, comprising the earliest sequence generated from the country (InDRE04. GenBank Accession KP795107 [67]), and a cluster of sequences from the southwestern state of Chiapas [54] (named here clade A; posterior probability, PP = 0.88) (**Fig 2B**). While sequences within clade A were generally no longer sampled after late 2014, a second well-supported clade (named here clade B; posterior probability, PP = 0.91) was identified. Before being introduced into Mexico, our results indicate that clade B had initially circulated within the Eastern Caribbean, but then shifted towards the Central Caribbean. Introductions into the USA occurred multiple times from both the Eastern and Central Caribbean regions, leading to subsequent introductions into Nicaragua (**S5 Fig**). Within clade B, Mexico virus genomes directly descend from sequences sampled from Nicaragua and the USA. Nonetheless, given the overrepresentation of sequences from the USA and Nicaragua (**S2B Fig**), it is likely that sampling biases may partially account for Nicaragua as an inferred source location.

The reconstruction of virus introduction events of CHIKV into Mexico revealed 13 clusters with MRCAs likely representing independent introductions from abroad, confirming previous observations on multiple independent introductions observed for CHIKV in Mexico [55,67–69]. In seven instances, Mexico sequences represent singleton events, denoting either returning travelers (as is the case with the aforementioned InDRE04 sequence), or poorly sampled transmission chains (*i.e.*, clades). The remaining six genomes fall within six well-supported clusters exclusive to Mexico (named here *MX_cluster_1—MX_cluster_6*). For these clusters, ancestral MRCA nodes were inferred to have emerged within the Southwest ($PP_{MX\_cluster\_1}$ = 0.5619, $P_{Southwest}$ = 0.7649; $PP_{MX\_cluster\_3}$ = 0.9998, $P_{Southwest}$ = 0.662; $PP_{MX\_cluster\_4}$ = 0.9462, $P_{Southwest}$ = 0.8362; $PP_{MX\_cluster\_6}$ = 0.9971, $P_{Southwest}$ = 0.9994), in the East ($PP_{MX\_cluster\_2}$ = 0.5364, $P_{East}$ = 0.6846) and in the West of the country ($PP_{MX\_cluster\_5}$ = 0.5047, $P_{West}$ = 0.9092) (**Fig 2B**). We identified three significant transition rates across locations between the Americas and Mexico, namely into the Southwest (BF = 5236.52; PP > 0.99), Centre-north (BF = 86.21; PP = 0.92) and West (BF = 15.76; PP = 0.68) regions (Fig 2B and 2C). The MRCAs for the six Mexico clusters were all dated between mid- and late-2014 (Table 2), with two of these

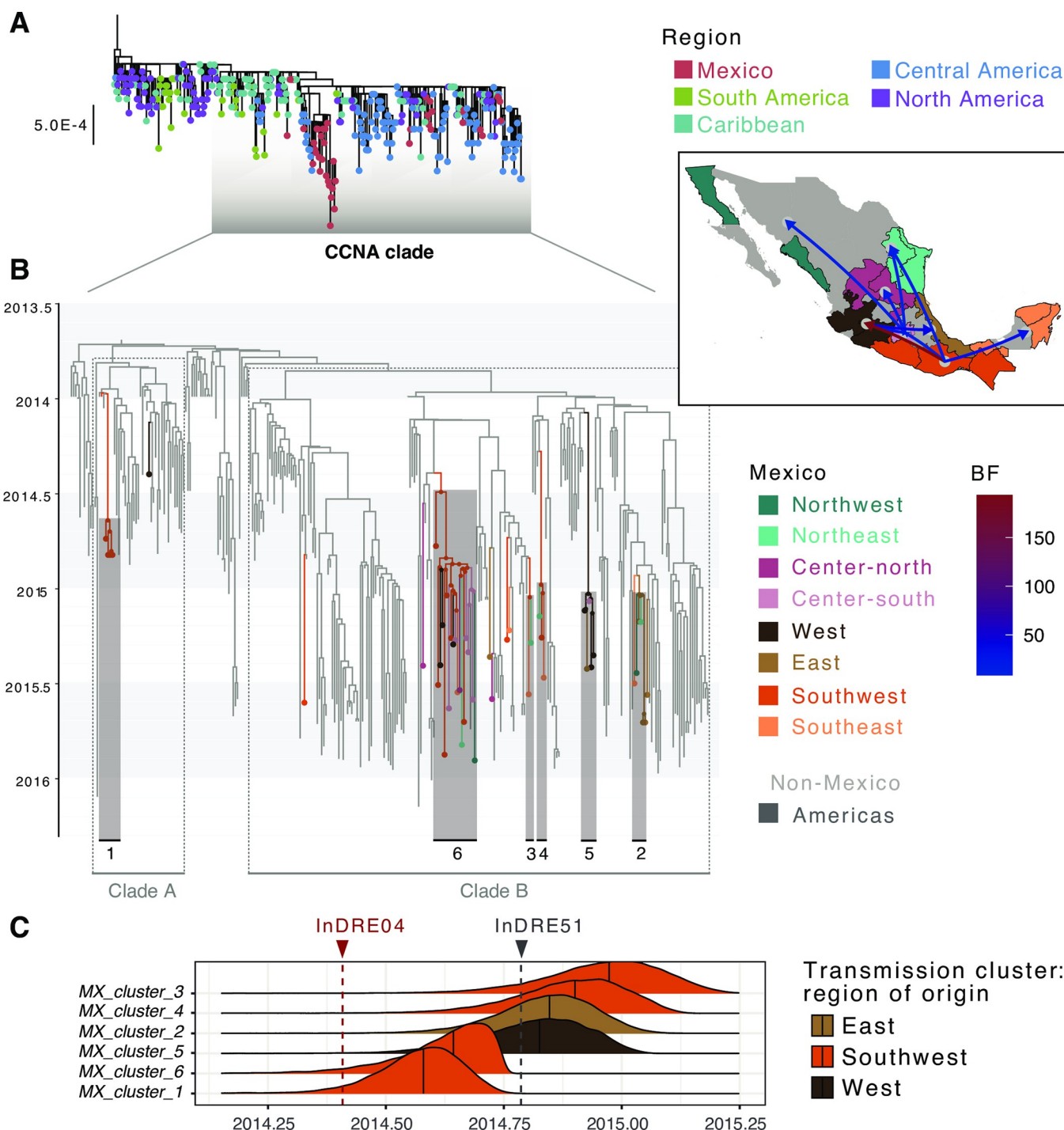

**Fig 2. Time-scaled analysis for CHIKV in Mexico. (A)** ML phylogenetic tree of all CHIKV complete genome sequences from the Americas included in our analysis, with tips coloured by region of collection. **(B)** Time-calibrated phylogeographic analysis of the CHIKV CCNA. Tip and nodes for locations within Mexico and are coloured by region, with the main clusters identified numbered (1–6). The map in the inset shows pairs of locations where transition rates significantly explain the phylogenetic diffusion process, as inferred under a BSSVS analysis. Only transition rates with a posterior probability (PP) > 0.5 are shown, coloured by Bayes Factor (BF). **(C)** Posterior probability densities for the TMRCAs of six CHIKV transmission clusters in Mexico. Median values for each distribution are indicated, with the distributions coloured by the most probable location of each MRCA. Collection dates for InDRE04 (the first sequence corresponding to a CHIKV imported case into Mexico, collected on 2014-05-30), and InDRE51 (the first confirmed autochthonous case within Mexico, collected on 2014-10-15) are shown in red and grey, respectively. Plots were generated using the ggplot package (https://ggplot2.tidyverse.org/index.html) in R. Original base layer maps use as a source for geospatial base layer data public domain maps imported from the Natural Earth data project (https://cran.r-project.org/web/packages/rnaturalearth/index.html).

**Table 2. Clusters inferred from phylogeographic analyses.**

| Virus | | Mexico cluster | TMRCA | | Ancestral location |
|---|---|---|---|---|---|
| | | | *Median age* | *95% HPD* | |
| *CHIKV* | | *MX_cluster_1* | 2014.57 | 2013.69–2014.74 | Southwest |
| | | *MX_cluster_6* | 2014.64 | 2014.42–2014.75 | Southwest |
| | | *MX_cluster_5* | 2014.83 | 2014.59–2015.01 | East |
| | | *MX_cluster_2* | 2014.85 | 2014.62–2014.04 | West |
| | | *MX_Cluster_4* | 2014.9 | 2014.6–2015.11 | Southwest |
| | | *MX_cluster_3* | 2014.97 | 2014.72–2015.18 | Southwest |
| *DENV* | DENV-2 | *cluster_2* | 1998.97 | 1997.35–2000.61 | Southwest |
| | DENV-2 | *cluster_1* | 1999.51 | 1998.13–2000.79 | Southwest |
| | DENV-1 | *cluster_2* | 2003.49 | 2002.31–2004.18 | – |
| | DENV-1 | *cluster_1* | 2003.81 | 2001.63–2003.91 | – |
| | DENV-1 | *cluster_3* | 2004.03 | 2003.01–2004.8 | – |
| | DENV-2 | *cluster_3* | 2008.02 | 2007.15–2008.9 | Southwest |
| | DENV-1 | *cluster_4* | 2009.35 | 2008.28–2010.2 | – |

(*MX_cluster_1* and *MX_cluster_6*) predating the first autochthonous CHIKV case officially reported in the country (represented by sequence InDRE51, GenBank accession number KP795109). However, inferred dates for these MRCAs are later than the sampling date of the earliest reported introduced CHIKV case in Mexico (InDRE04), with a known a travel history into the Caribbean [67] (**Fig 2B**). The remaining clusters circulated in the country since October 2014 (**Table 2**).

## Spatial dynamics of CHIKV across Central America and Mexico

The spread of CHIKV within Mexico was inferred to occur from the Southwest across the southern and central coastal regions of Mexico. We identified 14 supported transitions, with source regions predominantly represented by the West, Southwest and Centre-south of the country (**Fig 2A** and **S3 Table**). The *MX_cluster_1* makes up the largest transmission chain identified for CHIKV, circulating across 11 different states (Baja California Norte, Mexico state, Mexico City, Guerrero, Tamaulipas, San Luis Potosí, Quintana Roo, Jalisco, Morelos, Michoacán and Oaxaca). The second largest cluster (*MX_cluster_6)*, corresponds to the earliest group of cases sampled from the state of Chiapas, and is exclusive to the Southwest region [54]. Contrastingly, *MX_cluster_3 and MX_cluster_4* were only sampled from two regions each (comprising only 3 and 2 sequences, respectively), whilst *MX_cluster_5* circulated only across two regions in central Mexico (East and West). Finally, *MX_cluster_2* circulated across three regions in central and northern Mexico (East, Northeast and Northwest).

## DENV genome data

We were able to generate 7 complete DENV-1 and 11 DENV-2 partial genomes. DENV genome sequences generated have collection dates between 2013 and 2017 with a geographical representation of up to six different states (**S1 Table**). For DENV-2, insufficient coverage for several genomic regions (including both UTRs) was observed due to a partial failure during the multiplex PCR (genome regions that failed to amplify were masked with Ns (**S3 Data**). Nonetheless, successfully sequenced sites (5104/10177 bases, corresponding to ~50% of the genome length) encompassed partial ORFs for the capsid, envelope and glycoprotein (as well as various non-structural protein partial gene regions: NS1, NS2A, NS2B, NS3, NS4B and NS5). Thus, the virus genome regions

successfully sequenced provided enough phylogenetic signal for further analyses, further highlighting the potential use of incomplete genome data for genomic epidemiologic-based phylo-dynamic inference [70]. Again, newly generated genome data corresponding to DENV in Mexico represent a considerable improvement of the spatiotemporal representation, which previously represented only genome sequences sampled from 2012. Furthermore, genomes now represent two states with no previous data for DENV-1 (Chiapas and Veracruz), and four previously unrepresented states for DENV-2 (Colima, Jalisco, Morelos and Veracruz) (S1 Table).

## Epidemiological dynamics of DENV 1–2 in Mexico from 2013–2019

Analysis of DENV longitudinal epidemiological data from Mexico shows a clear virus seasonal pattern fluctuating between 2013 and 2019, with distinct trends across regions (S6 Fig). During 2013, the Northeast, East and Southeast regions reported up to 3000 monthly cases at their respective peaks, with similar case numbers observed in the Northwest, West and Southwest regions, (with monthly cases not exceeding 2500) (S6 Fig). The proportion of serotyped cases increased substantially between 2017 and 2018, when the yearly cumulative number of cases was at the lowest (S6 Fig). Most samples were identified as DENV-1 or DENV-2, with a smaller portion of cases identified as DENV-3 or co-infections with multiple serotypes (Fig 3). When our results were contextualised alongside previously reported epidemiological data available for DENV serotypes from 2000 to 2013 [71], it becomes clear that DENV-1 was practically absent from the country until 2004 and 2007, when it began to dominate over other viral serotypes. Since then, DENV-1 and DENV-2 have been co-circulating showing epidemiological dominance (Fig 3A). DENV serotype replacements have been widely described in Asia [72–74] and South America [75], whilst a few studies of DENV in Mexico report frequent lineage replacement events observed for distinct serotypes [31,32]. The periodic importations of DENV into Mexico we detect throughout 2000–2010 may (partially) explain the shifting dominance observed for specific DENV virus serotypes across time and space.

For the DENV-1 data, a remarkable anomaly was noted: a peak in the number of DENV-1 cases was observed during 2017, whilst an equal proportion of both DENV-1 and DENV-2 serotypes was recorded at the same time (Fig 3B). This observation can be (partially) explained by the combination of an overall lower number of DENV cases in the country during this season, coupled with a highest rate of serotyping and case reporting from the Centre-north region of the country (S7 Fig). Specifically, during 2017, the proportion of serotyped samples increased in relation to previous years (S8 Fig), with the Centre-north region serotyping ~95% of all cases. At the same time, the majority of cases typified as DENV-1 also came from the Centre-north, resulting in a biased epidemiological trend (S7–S9 Figs). Thus, when breaking down epidemiological patterns per region, some common trends were observed (Fig 3C). Samples classified as 'other' DENV serotypes (neither DENV-1 or DENV-2, mainly corresponding to DENV-3 and 4) comprised the minority of the total serotyped cases, and were prominently observed in the East and Northeast region in 2016 (but were completely absent from the North-western region). The case proportion of DENV-2 in relation to DENV-1 decreased between 2013 and 2016 for most of the country, except for the Southeast region. Then, between 2016 and 2017, DENV-2 increased in frequency across the Northeast and Southwest regions, followed by the rest of the country (particularly in the West and Northwest) as later observed throughout 2018 (Fig 3C).

## Evolutionary dynamics of DENV-1 and DENV-2 in Central America and Mexico

Compared to CHIKV, our phylogeographic analyses revealed that DENV-1 and DENV-2 display similar epidemiological and evolutionary trends. Specifically, for both DENV trees,

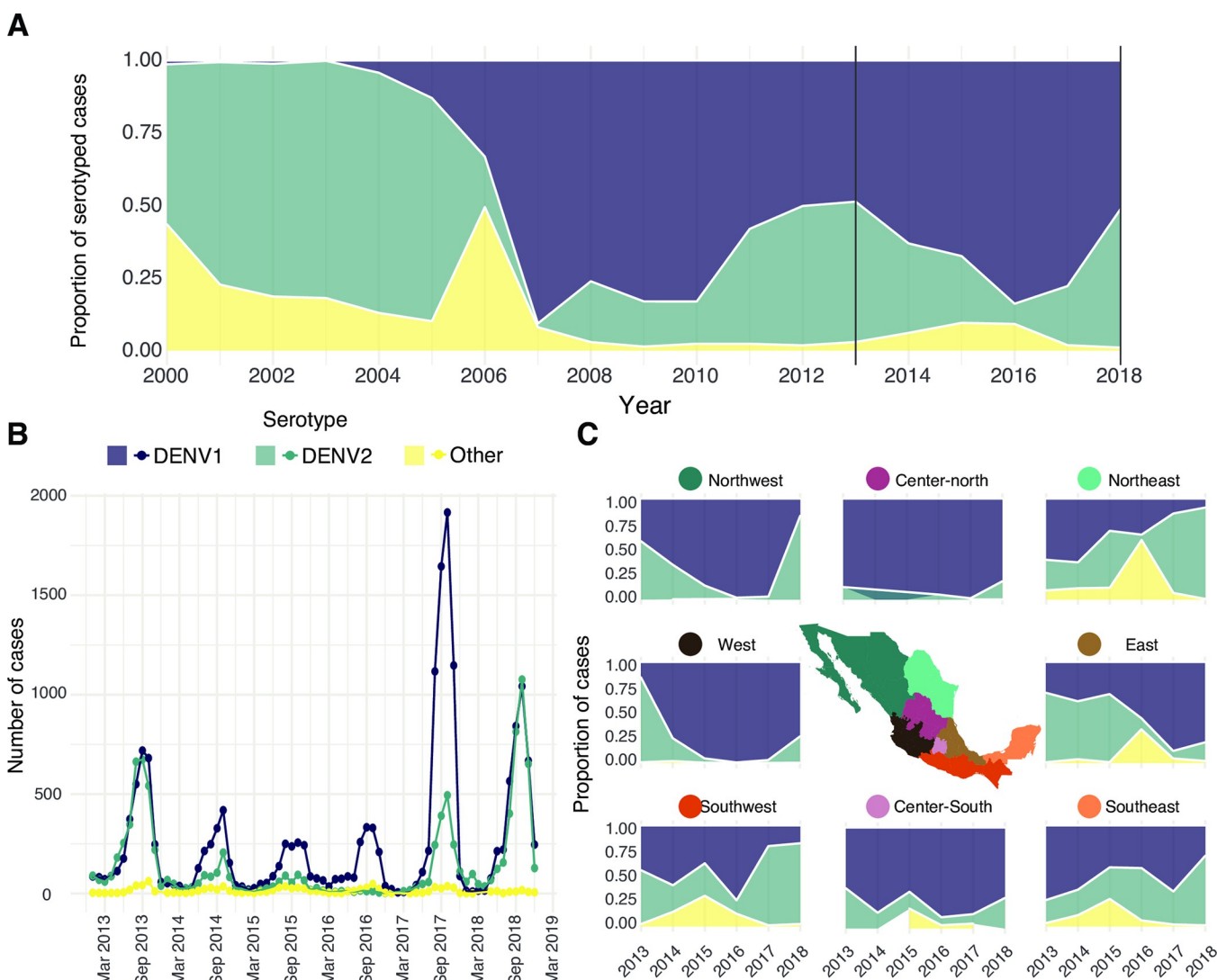

**Fig 3. Epidemiological trends of DENV in Mexico from 2013 to 2019. (A)** The proportion of serotyped DENV cases between 2000 and 2018 in Mexico shown in reference to the total number of serotyped samples per year for all the country. Data provided by InDRE for this study is shown between the solid black lines (delimiting 2013 to 2018). Data from previous years was obtained from published sources [71]. The DENV-3 and DENV-4 serotypes together with potential co-infections (reported between 2013 and 2018) are grouped into the 'Other' category. **(B)** Monthly number of cases assigned to each DENV serotype in Mexico from 2013 to 2018. **(C)** Breakdown of serotype proportions in different geographic regions in Mexico between 2013 and 2019, relative to the total number of serotyped samples for each location. Plots were generated using the ggplot package (https://ggplot2.tidyverse.org/index.html) in R. Original base layer maps use as a source for geospatial base layer data public domain maps imported from the Natural Earth data project (https://cran.r-project.org/web/packages/rnaturalearth/index.html).

sequences from Mexico and Nicaragua group together into single clades, along with a few sequences from Central American, North American and/or the Caribbean in both cases (such clades are henceforth named here as CAM) (**S10 Fig**). The CAM clades are comparable to the CCNA clade described for CHIKV, but were generally found to be geographically constrained. For DENV-1, the CAM clade dates back to the early 2000s (median age MRCA $_{DENV-1-CAM}$ = 2000.5152, 95% HPD = 1997.8814–2002.2687), whilst for DENV-2 to the late 1990s (median age MRCA $_{DENV-2-CAM}$ = 1996.8204, 95% HPD = 1995.6215–1997.7913). Reconstructing the importation and spread patterns of DENV-1 and 2 in Mexico revealed that both CAM clades circulated predominantly in Nicaragua before they were periodically and independently

introduced into Mexico. Five different introductions were identified for DENV-1, whilst three were detected for DENV-2. Nonetheless, as highlighted before, the overrepresentation of DENV genome sequences from Nicaragua, together with a limited reduced genome representation from other countries in the region, are likely to impact phylogeographic reconstructions (**S3 Table**).

The sampling timespan for DENV viruses Mexico is mostly overlapping, with DENV-1 sequences collected between 2004 and 2017, and DENV-2 sequences collected between 2000 and 2019. The MRCAs for different clusters within the CAM clades show that both virus subtypes circulated within the country since the late 1990s. For DENV-2, two clusters date back to 1999 and circulated until 2013, displaying a persistence of approximately 14 years. Consistently, for DENV-1, three clusters date back to the early 2000s, also circulating up to 2013. Later DENV introductions are associated with a cluster dating back to 2008 (for DENV-2), and another cluster dating back to 2009 (for DENV-1) (**S10 Fig** and **S3 Table**).

### Spatial dynamics of DENV-2 across Central America and Mexico

Whilst DENV-2 genomes have been sampled extensively across southern and central Mexico, the geographical distribution for DENV-1 genomes is mostly limited to the Southeast region. Thus, DENV-2 is used here to illustrate general DENV spread dynamics across the country (**Fig 4**).

Phylogeographic reconstruction shows that the three DENV-2 clusters are likely to have originated in the Southwest of the country, further spreading towards the East and West coasts, into the Yucatán peninsula (**Fig 4**). Movements observed between Nicaragua and the Southwest of the country support the importation events inferred for DENV-2 into Mexico (BF = 1124.01, PP = 0.99), with subsequent spread into other states towards the North. Within Mexico, viral movements from the Southwest into the West (BF = 2929.53, PP > 0.99), Southeast (BF = 1270.88, PP > 0.99) and East (BF = 56.08, PP = 0.90) were inferred as significant. Movements from the West to the Centre-south region (namely the state of Morelos) were also significant (BF = 83.25, PP = 0.93), as well as across border movements from south into Guatemala (BF = 134.65, PP = 0.96).

For DENV-1 and DENV-2, given the availability of longitudinally-collected epidemiological data for over almost two decades within the country, changes in the population size over time can be estimated by integrating genomic data under a Bayesian Skyline plot (BSP) (see Methods section: *Time-calibrated phylogeographic analyses)*. The BSPs for both the DENV-1 and DENV-2 CAM lineages show different periods of time representing viral population growth (suggesting an increase in the number of new cases in Mexico): 2005–2007 and 2011–2013 for DENV-1, and 2017–2018 for DENV-2 (**S11 Fig**). These periods coincide with a relative increase in cases associated to each specific serotype: for DENV-1 observed between 2005–2007, and for DENV-2 between 2017–2018. For DENV-2, the population growth period identified between 2011–2013 overlaps with a decrease in the serotype's proportion relative to DENV-1, followed a subsequent increase observed between 2013–2016. Overall, no clear pattern between the BSP and serotype frequencies was observed.

### Discussion

In order to investigate convergence across the epidemiological and evolutionary dynamics of CHIKV, DENV-1 and DENV-2 viruses circulating in Mexico, we generated complete and partial virus genomes, and analysed them together with genomic and epidemiological data from the Americas available for over almost two decades. We find important similarities represented by multiple virus introduction events into the country derived from lineages predominantly

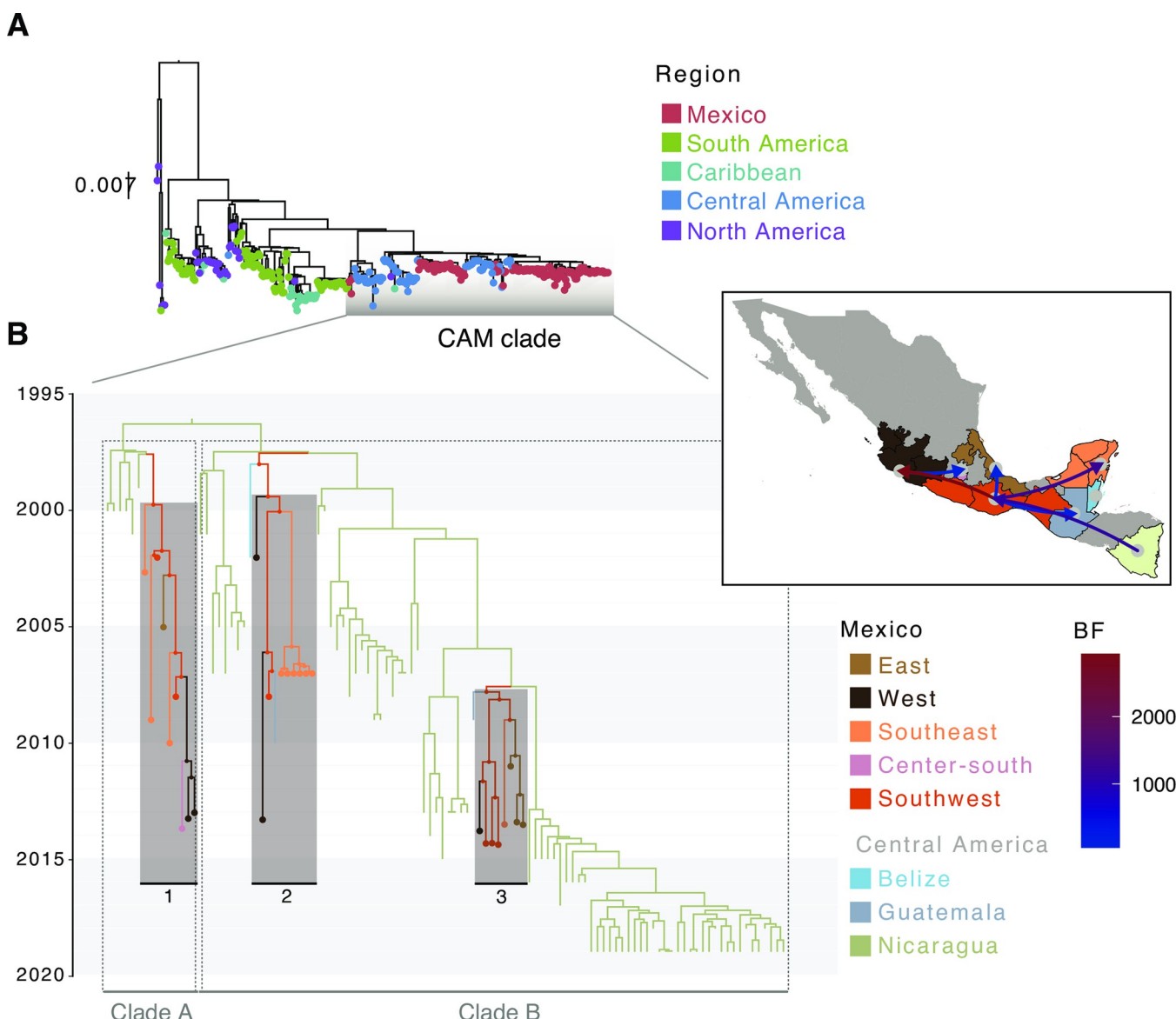

**Fig 4. Time-scaled analysis for DENV-2 in Mexico.** **(A)** ML phylogenetic tree for DENV-2 in the Americas, with tips coloured according to the geographic region of collection. **(B)** Time-calibrated phylogeographic analysis for the DENV-2 CAM lineage, with the MCC tree displaying different sampling locations for Mexico. Tip and nodes shapes are included for locations within Mexico, coloured by region. Distinct clades identified within the CAM lineage are designated as Clade A (including one cluster from Mexico) and Clade B (including two clusters from Mexico). The map in the inset shows pairs of locations for which transition rates were inferred to be significant under a BSSVS analysis. Only transition rates with a posterior probability (PP) > 0.5 are shown, coloured by Bayes Factor (BF). Original base layer maps use as a source for geospatial base layer data public domain maps imported from the Natural Earth data project (https://cran.r-project.org/web/packages/rnaturalearth/index.html).

sampled from the Caribbean and Central America. These three Aedes-borne arboviruses also share comparable spatiotemporal transmission trends, in which the Southwest region of the country seems to have played a pivotal role in virus seeding and spread across Mexico.

The general spatial dynamics derived from phylogeographic analysis for DENV in Mexico (represented by DENV-2) revealed multiple virus importations into the country leading to extended transmission chains, with a most probable within-country location for viral introductions determined to be the state of Chiapas. Comprising a better spatial and temporal

sampling within a single epidemic, the spatial diffusion pattern observed for CHIKV was also found to be comparable, with most transmission clusters displaying the same inferred ancestral location (Chiapas). Of notice, in an independent study by Thézé et al. 2018 [70], a similar trend was also described for ZIKV in Mexico, again showing a most likely virus introduction into Chiapas, with subsequent spread towards the north of the country. Our results further support for an unbiased inference, as viral genomes sampled from this region were not particularly overrepresented within our datasets. However, an extended genomic representation within the country could contribute to portray a more complete panorama of virus spread patterns at a higher resolution (*i.e.*, across states).

Despite apparent contrasting epidemiological trends observed among emerging and established Aedes-borne arboviruses (e.g. DENV vs CHIKV), it is expected that emerging viruses may, at some point, display seasonal patterns coupled with a prolonged regional endemicity [76]. This is clear in countries like India and Thailand, where CHIKV has circulated for an extended period of time. In such regions, cases tend to peak primarily during the rainy season, given the favourable environmental conditions that contribute to the breeding and proliferation of Aedes mosquitoes [77,78]. In the Americas, a similar pattern was already observed during the initial CHIKV outbreak in the Caribbean [76]. Subsequently, in Brazil, where the CHIKV has circulated for longer, seasonality is evident, with an increased virus transmission over time occurring from December to March in most parts of the country [53,79]. In Mexico, only a handful of new CHIKV cases have been officially reported after 2018 [80]. However, serological surveys across different southern states denote a significant disease incidence and prevalence [81], suggesting that there is substantial underreporting of cases nationwide. Thus, we speculate that in Mexico, CHIKV is likely to circulate cryptically, where a prolonged virus endemicity could exhibit seasonal epidemiological trends (similar to those historically observed for DENV). However, only an increased targeted surveillance in high-risk states (comprising Chiapas, Oaxaca, Guerrero, Veracruz, Tamaulipas, Quintana Roo, Campeche and Yucatán, denoting those historically most affected by Aedes-borne viruses) could shed light on the cryptic epidemiological behaviour of CHIKV, as well as other emerging and established Aedes-borne arboviruses circulating in Mexico.

Within a global context, southern tropical regions are known to provide the ideal ecological conditions related to vector competence and establishment, and thus have been historically associated with Aedes-borne arboviral outbreaks [4,74]. In the Americas, there is evidence that Aedes-borne arboviruses display seeding and spread trends that derive from high-prevalence areas [82]. In this light, longitudinal epidemiological case data derived from arbovirus epidemics in Mexico support for the southern tropical region of the country as an important hub for viral transmission [70], regardless of putative systematic biases towards an increased case detection given an intensified surveillance within the area. Thus, the independent association of the Southwestern region of Mexico as a source location for the introduction of DENV, CHIKV and ZIKV, provides strong evidence for convergence in the spatial dynamics of both seasonal and emerging Aedes-borne arboviruses circulating within the country.

The spread of Aedes-borne arboviruses in the Americas is largely known to be driven by human mobility patterns, as mosquitoes tend to follow human populations [24]. In agreement, our results suggest that Aedes-borne arboviral importation and spread in Mexico has been driven (to some extent) by human movements, as has been observed in neighbouring countries [4,74]. Furthermore, some of the virus diffusion patterns we observe are potentially linked to human movements across borders, particularly following land-based unregulated migration routes from central America into Mexico and towards the USA [83–86]. Thus, a migration-informed genomic epidemiology-based disease surveillance could contribute to virus control

strategies, as human mobility is expected to potentially increase due to changing climatic, political and socio-economic factors.

Nonetheless, whilst human mobility is an important determinant for arbovirus epidemics, viral spread is very much shaped by other climatic, ecological and immunological factors [1,9–11,87]. As an example, the ability of Aedes-borne arboviruses to establish a sustained or seasonal transmission in a recipient geographical area, depends on climatic and ecological condition, as well as the viral capacity to persistently infect the vector population. In Mexico, arbovirus persistence is reflected by somewhat independent epidemiological trends observed across distinct regions of the country. This may be exemplified by the Yucatan peninsula, that compared to other regions of Mexico, showed a unique epidemiological trend during the late CHIKV peak in 2015. Nonetheless, exploring arbovirus spatial diffusion patterns across different scales can only be achieved through comprehensive approaches that integrate multiple sourced information (including social, ecological, genomic and epidemiological data).

Predicting spatial spread patterns for Aedes-borne arboviruses could play an important role in the design of surveillance programs and public health interventions, such as the implementation of vaccine trials. In this scenario, previous immunity within the host population is an important covariate to consider when interpreting our results. In this sense, it is not a surprise that the Southwest of the country (displaying the ideal ecological suitability for vectors, coupled with a higher probability of new arboviral importations that may result in extended transmission chains) shows the highest seroprevalence for DENV [11]. Congruently, other regions in Mexico showing subsequent domestic virus importations from the Southwest (such as the West and Southeast), also show a high seroprevalence [88,89]. Thus, our results may inform on suitable locations for vaccine trials targeting multiple Aedes-borne arboviruses applicable to naïve individuals, where high numbers of new cases are required to achieve targets of efficacy promptly [90].

Limitations of our study include biases for inferred source locations for arboviral introductions from abroad, likely to be impacted by differences in genomic surveillance efforts across countries within the same region. In this light, multiple viral introductions were inferred from Nicaragua into the country, indicating that arbovirus epidemics in Mexico are likely affected by neighbouring regions [91]. Nevertheless, whilst Nicaragua features a considerable genomic representation in Latin America [37,38], other countries in central America (such as Belize, Guatemala, Honduras and El Salvador) have limited publicly available arbovirus genome data, and thus virus introductions from these cannot be inferred. Further limitations comprise sampling gaps that can bias the inference of ancestral locations at nodes [92], reflected by long branches separating the clusters identified (indicative of long cryptic circulation periods, or an unsampled virus genetic diversity) that may result in uncertainty in date estimates derived from time-calibrated phylogenies. Thus, only an enhanced spatial and temporal virus genome sampling and sequencing across bordering countries will enable a more detailed exploration of arboviral dynamics in the region.

As evidenced for CHIKV, the cumulative number of virus genomes by country does not correlate with officially reported case numbers, and this is likely to mirror data for other Aedes-borne arboviruses in the Americas. Heterogeneous data sampling prevents a comprehensive outbreak characterization, further accentuating the need to strengthen genomic epidemiology-based surveillance across regions. This also denotes regional disparities in the intensity of virus genome sampling and sequencing through time. As an example of this, for some countries like Brazil, an extensive sampling of ZIKV enabled exploring the spatial and temporal dynamics of the virus across the country [6,93], whereas the lack of data from neighbouring countries left substantial knowledge gaps in the global overview of the epidemic within the Americas. Despite a limited virus sequencing for Aedes-borne arboviruses in Mexico, we were still able successfully quantify and characterize lineage-specific transmission

chains across the country. Nevertheless, a more uniform sampling could uncover additional clades, offer finer insights into the geographic distribution of such clades, and enhance viral spread reconstructions at a higher resolution.

Multiple tools have been developed to guide the implementation of interventions to control the transmission of arboviral diseases [82,91,94]. Our results add to the current knowledge of arboviral dynamics in Mexico, highlighting the potential predictability of spatial invasion dynamics. Further incorporating our results into current epidemiological and spatial models may enhance the accuracy of current risk maps for established and emerging Aedes-borne arboviruses in Mexico and Central America. Finally, the key role of neighbouring Latin American countries in the development of arboviral epidemics in Mexico, and of the country's southern border in the spread of Aedes-borne arboviruses at national scale, prompts the need to better understand the role of anthropogenic factors in the transmission dynamics of viral pathogens, particularly concerning the effects of land-based migration. Our study further pinpoints on how joint efforts between public health and academic institutions can foster genomic epidemiology-based surveillance strategies applied to the developing world.

## Supporting information

**S1 Text. Retrospective on Aedes-borne arbovirus epidemics in the Americas and epidemiological surveillance in Mexico.**
(DOCX)

**S1 Fig. Geographic distribution of Aedes-borne arbovirus genome sequences sampled for phylogeographic analyses. (A)** Geographic distribution of DENV-1, DENV-2 and CHIKV virus genomes from the region of the Americas included for phylogeographic analyses in this study. Maps are coloured according to the number of sequences sampled per country. **(B)** Geographic distribution of DENV-1, DENV-2 and CHIKV virus genomes from Mexico included in this study. The map on left indicates the 32 states from the country, whilst the map of the right shows in different colours the distinct geographic regions (comprising different states) from which Aedes-borne virus genomes are available from. Plots were generated using the ggplot package (https://ggplot2.tidyverse.org/index.html) in R. Original base layer maps use as a source for geospatial base layer data public domain maps imported from the Natural Earth data project (https://cran.r-project.org/web/packages/rnaturalearth/index.html).
(PNG)

**S2 Fig. CHIKV epidemiological and genomic surveillance trends in the Americas. (A)** Monthly number of CHIKV cases reported to the Pan-American Health Organisation (PAHO) between 2013 and 2017, grouped by country. **(B)** Comparison between monthly number of cases (reported to PAHO, upper panel) in countries that have generated CHIKV genome sequences, and publicly available complete CHIKV genome sequences (lower panel). Mexico sequences include those generated in this study. Plots were generated using the ggplot package (https://ggplot2.tidyverse.org/index.html) in R.
(PNG)

**S3 Fig. Complete CHIKV genome sequences versus total cases reported to PAHO per country. Linear regression between the numbers of CHIKV sequences and cases per country over time, with 95% confidence interval (CI) shown in grey.** A Spearman's Rho = 0.26, $p$ = 0.15 denotes no correlation between the cumulative number of cases per country versus the number of viral genome sequences available per country. Plots were generated using the ggplot package (https://ggplot2.tidyverse.org/index.html) in R.
(PNG)

**S4 Fig. CHIKV epidemiological trends across Mexico between 2016 and 2018.** Monthly cases for CHIKV reported under the SINAVE surveillance system (InDRE/Ministry of Health Mexico). Plots were generated using the ggplot package (https://ggplot2.tidyverse.org/index.html) in R.
(PNG)

**S5 Fig. Phylogenetic analyses of CHIKV in the Americas.** ML phylogenetic tree for CHIKV inferred from the complete genome sequences from the Americas included in our analysis, denoting an 'American' lineage (left panel). Tree tips are coloured according to the region/country of collection, whilst nodes are coloured according to branch support values (SH-aLRT). To the right, a time-calibrated MCC tree is displayed, showing a well-defined CCNA clade within the 'American' lineage. Tips and branches are coloured by the location of origin and circulation, inferred through a DTA phylogeographic analysis (see Methods section, main text).
(PNG)

**S6 Fig. DENV epidemiological trends across Mexico between 2016 and 2018.** Monthly cases for DENV (aggregating both dengue fever and dengue haemorrhagic fever) reported under the SINAVE surveillance system (InDRE/Ministry of Health Mexico). Plots were generated using the ggplot package (https://ggplot2.tidyverse.org/index.html) in R.
(PNG)

**S7 Fig. Serotyping representation for DENV across regions in Mexico.** Total number of DENV cases reported by year (black line), compared to total number of DENV cases where the causal serotype has been identified and reported (grey line). Data corresponds to the period of time between 2013 and 2018, as reported under the SINAVE surveillance system (InDRE/Ministry of Health Mexico). Plots were generated using the ggplot package (https://ggplot2.tidyverse.org/index.html) in R.
(PNG)

**S8 Fig. DENV serotyping efficacy across time in different Mexico regions.** Percentage of serotyped samples per region per year between 2013 and 2018. Plots were generated using the ggplot package (https://ggplot2.tidyverse.org/index.html) in R.
(PNG)

**S9 Fig. DENV-1 and DENV-2 case numbers across Mexico regions.** Monthly numbers of cases identified as DENV-1 (purple) or DENV-2 (teal) between 2013 and 2018 across geographic regions in the country. Plots were generated using the ggplot package (https://ggplot2.tidyverse.org/index.html) in R.
(PNG)

**S10 Fig. Phylogenetic analyses of DENV-1 and DENV-2 in the Americas.** ML phylogenetic trees for DENV-1 (above) and for DENV-2 (below) inferred from the complete genome sequences from the Americas included in our analysis are shown to the left. Tree tips are coloured according to the country/region of collection, whilst nodes are coloured according to branch support values (SH-aLRT). To the right, the time-calibrated MCC trees are displayed, showing well-defined CCNA clades for each virus. Tips and branches are coloured by the location of origin and circulation, inferred through a DTA phylogeographic analysis (see Methods section, main text).
(PNG)

**S11 Fig. Bayesian Skyline plot of the CAM DENV-1 and DENV-2 lineages.** Upper and lower panels show the Bayesian Skyline plots (BSPs) obtained from the time-scaled phylogenetic analyses of the DENV-1 and DENV-2 CCNA lineages. The middle panel shows the proportion of serotyped cases for each DENV serotype in Mexico over a comparable period of time. Shading in purple (for DENV-1) and green (for DENV-2) show periods of time where an increase in the virus effective population size over time was observed (as suggested by the BSP), highlighting the proportion of virus serotypes.
(PNG)

**S1 Table. Genome sequences generated in this study.**
(XLSX)

**S2 Table. PAHO regions.**
(XLSX)

**S3 Table. BSSVS results for CHIKV.**
(XLSX)

**S4 Table. BSSVS results for DENV-2.**
(XLSX)

**S1 Data. Alignment for CHIKV virus genome sequences (can be opened with any simple text editor for.txt, or with any sequence alignment visualization software, e.g., AliView, MEGA, Geneious).**
(FASTA)

**S2 Data. Alignment for DENV-1 virus genome sequences (can be opened with any simple text editor for.txt, or with any sequence alignment visualization software, e.g., AliView, MEGA, Geneious).**
(FASTA)

**S3 Data. Alignment for DENV-2 virus genome sequences (can be opened with any simple text editor for.txt, or with any sequence alignment visualization software, e.g., AliView, MEGA, Geneious).**
(FASTA)

**S4 Data. Accession numbers for virus genome sequences used in this study.**
(XLSX)

## Acknowledgments

We would like to thank the personnel at the National Institute for Epidemiological Diagnosis and Reference (Instituto de Diagnóstico y Referencia Epidemiológicos "Dr. Manuel Martínez Báez"–InDRE, Secretaría de Salud, Mexico- WHO Collaborating Center in Arbovirus), for their support in sharing samples and data from the 'National Arbovirus Reference Laboratory', and for contributing to the molecular biology and sequencing laboratory work at the 'Technological development and molecular research unit'. We thank Dr. Santa Elizabeth Ceballos Liceaga, Head of the Epidemiological Surveillance of Communicable Diseases at the 'General Directorate of Epidemiology' (DGE, Secretaría de Salud, Mexico) for her support in sharing epidemiological data. We also thank Dr. Lorena Preciado-Llanes from the Jenner Institute at the University of Oxford for her assistance with funds management and reporting. We would also like to thank Seth Flaxman and Simon Dellicour for their input and insightful discussions on this manuscript.

## Author Contributions

**Conceptualization:** Marina Escalera-Zamudio.

**Data curation:** Bernardo Gutierrez, Darlan da Silva Candido, Sumali Bajaj, Sarah C. Hill, Julien Thézé.

**Formal analysis:** Bernardo Gutierrez, Darlan da Silva Candido, Sumali Bajaj.

**Funding acquisition:** Oliver G. Pybus, Arturo Reyes-Sandoval, Moritz U. G. Kraemer, Marina Escalera-Zamudio.

**Investigation:** Bernardo Gutierrez, Darlan da Silva Candido, Sumali Bajaj, Sarah C. Hill, Oliver G. Pybus, Arturo Reyes-Sandoval, Moritz U. G. Kraemer, Marina Escalera-Zamudio.

**Methodology:** Bernardo Gutierrez, Darlan da Silva Candido, Abril Paulina Rodriguez Maldonado, Fabiola Garces Ayala, María de la Luz Torre Rodriguez, Adnan Araiza Rodriguez, Claudia Wong Arámbula, Mauricio Vázquez Pichardo.

**Project administration:** Ernesto Ramírez González, Irma López Martínez, José Alberto Díaz-Quiñónez, Mauricio Vázquez Pichardo, Oliver G. Pybus, Arturo Reyes-Sandoval, Moritz U. G. Kraemer, Marina Escalera-Zamudio.

**Resources:** Ernesto Ramírez González, Irma López Martínez, José Alberto Díaz-Quiñónez, Mauricio Vázquez Pichardo, Sarah C. Hill, Julien Thézé, Nuno R. Faria, Lorena Preciado-Llanes, Arturo Reyes-Sandoval, Moritz U. G. Kraemer.

**Software:** Sarah C. Hill, Julien Thézé.

**Supervision:** Ernesto Ramírez González, Irma López Martínez, José Alberto Díaz-Quiñónez, Mauricio Vázquez Pichardo, Nuno R. Faria, Oliver G. Pybus, Lorena Preciado-Llanes, Moritz U. G. Kraemer, Marina Escalera-Zamudio.

**Validation:** Oliver G. Pybus, Arturo Reyes-Sandoval, Marina Escalera-Zamudio.

**Visualization:** Marina Escalera-Zamudio.

**Writing – original draft:** Bernardo Gutierrez.

**Writing – review & editing:** Mauricio Vázquez Pichardo, Sarah C. Hill, Lorena Preciado-Llanes, Arturo Reyes-Sandoval, Moritz U. G. Kraemer, Marina Escalera-Zamudio.

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
