## [Decision Letter · Decision Letter 0]

14 May 2023

Dear Marina Escalera-Zamudio,

Thank you very much for submitting your manuscript "Convergent trends and spatiotemporal patterns of arboviruses in Mexico and Central America" for consideration at PLOS Neglected Tropical Diseases. As with all papers reviewed by the journal, your manuscript was reviewed by members of the editorial board and by several independent reviewers. The reviewers appreciated the attention to an important topic. Based on the reviews, we are likely to accept this manuscript for publication, providing that you modify the manuscript according to the review recommendations. 

Sincerely,

Aparna Krishnavajhala, Ph.D.

Academic Editor

Elvina Viennet

Section Editor

Reviewer's Responses to Questions

**Key Review Criteria Required for Acceptance?**

**Methods**

-Are the objectives of the study clearly articulated with a clear testable hypothesis stated?

-Is the study design appropriate to address the stated objectives?

-Is the population clearly described and appropriate for the hypothesis being tested?

-Is the sample size sufficient to ensure adequate power to address the hypothesis being tested?

-Were correct statistical analysis used to support conclusions?

-Are there concerns about ethical or regulatory requirements being met?

Reviewer #1: The objectives of the study are clearly stated, with appropriate study samples selected. More information on the selection process (if any are used) for samples sent to InDRE for long-term storage etc. would be useful and help to further contextualise the study samples. The methods used by the authors for CHIKV and DENV sequence generation, investigating epidemiological trends and phylogenetic and phytogeographic analyses are standard yet appropriate given the objectives, nature of the two viruses, as well as the scale of the study.

Reviewer #2: The study's objectives are clearly articulated and the hypothesis of the convergence in the epidemiology of Aedes-borne arboviruses in the Americas deserves future attention including the exploration of potential ecological factors underlying such convergence, as well as potential competition between the arboviruses sharing the same mosquito vectors. The approach that the authors have taken to address their objectives is valid and, importantly, they recognize and discuss the potential limitations and uncertainties capable of influencing the results of the study. The methods are sufficiently described and this section doesn't seem to need any improvements.

Reviewer #3: I have thoroughly gone through the manuscript. Authors have addressed the study very well. The key analysis are well described and used. All information clearly mentioned. Methodology and data are sufficient for the analysis.

**Results**

-Does the analysis presented match the analysis plan?

-Are the results clearly and completely presented?

-Are the figures (Tables, Images) of sufficient quality for clarity?

Reviewer #1: All results are clearly outlined and thoroughly presented in appropriate and detailed figures. The authors do a good job of presenting a breakdown of their results by region within Mexico, which is in line with the objectives and overall aims of the study. Though fewer than expected number of sequences were generated in this study (given its scale) the results do emphasise the difficulty in using historical samples for whole virus genome sequence generation, as well as highlight the lack of representation of these specific regions within Mexico with regard to these arboviruses. And even with the limited number of sequences generated (Table 1) the contribution of this study to data for this region is noteworthy. Furthermore, the authors do a commendable job of integrating their newly generated sequences with background sequences during their analyses and using the generated sequences regardless of having obtained less than 100% genome coverage. The latter point I find of particular importance given the recent interest in studying virus genomes that can be found in historical and stored samples, and the associated difficulty with complete genome retrieval.

Reviewer #2: While the results are presented clearly and completely there are several improvements to suggest:

- in the Methods sub-section (lines 119-121) describing phylogenetic analysis, there is a reference to the genomic dataset compiled for the analysis, which is provided as Supplementary Data 4. For better understanding of the geographical context and study results it would be good to include as a separate figure a map showing locations where the viruses were isolated; 

- the reference to Table 1 in the Methods section (line 157-158) seem to be an error, the mentions of the table in the Results seem correct.

Reviewer #3: Result portion is well written and inferred to the point from the analysis.

**Conclusions**

-Are the conclusions supported by the data presented?

-Are the limitations of analysis clearly described?

-Do the authors discuss how these data can be helpful to advance our understanding of the topic under study?

-Is public health relevance addressed?

Reviewer #1: The limitations of the methodology and analyses, in particular the phylogeographic analyses, are clearly outlined and accounted for within the Discussion section of the manuscript. The main findings are adequately compared with those from previous work and other work done in the region. Further discussion of the comparison of the findings, trends etc. identified for CHIKV and DENV in Mexico would strengthen the Discussion section.

Reviewer #2: The study conclusions are supported by the data presented and identifies important hotspots of viral introductions and spatial dissemination in the territory of Mexico, which is important for future surveillance and public health intervention efforts. The main limitations of the analysis are also sufficiently discussed along with potential applications of the results from this (and similar future) studies.

Reviewer #3: (No Response)

**Editorial and Data Presentation Modifications?**

Reviewer #1: A few minor grammatical errors to be corrected throughout the text.

Reviewer #2: 1) Some of the literature sources cited in the texts don't seem to be appropriate as they are not about arboviruses: i.e. references 2, 15 (Lassa fever virus), 3 (MERSV), 9(SARS-COV2), 16 (Rabies virus), 21 (Ebola virus). More appropriate sources pertaining to arboviruses and supporting the statements exist in the literature.

2) References 36-39 and 40-42, 83, 84 appear in the text not in the sequential order.

3) References 94-111 are not present in the main text. It looks like they are cited in Supplementary Text 1 but the supplementary should have its own list of references.

4) According to the journals' rules literature citations should be in square brackets in the text.

5) References 74, 75 seem irrelevant in the line 225. It would be better to cite the reference 36 there.

6) Abstract: The wording "Arboviruses cause both seasonal epidemics (e.g. dengue viruses, DENV) and emerging outbreaks (e.g. chikungunya and Zika viruses, CHIKV and ZIKV) with a significant impact on global health." Suggesting contrasting seasonal patterns for the studied viruses" is misleading because seasonality trends have been observed even at the height of CHIKV emergence in the Caribbean (e.g. see Johansson, 2015 Trends Parasitology) and with the prolonged endemicity in the Americas all of the viruses in question exhibit seasonal peaks. So, it's better to remove the viruses' names in the parentheses in the statement above.

7) It's better to refer to Aedes-borne arboviruses instead of just arboviruses throughout the text as ZIKV, CHIKV, and DENV are not fully representative of other arboviruses.

8) Some typos or tautologies are present throughout the text: "phylogenetic phylogenies" (line 138), "discrete phylogeographic discrete trait analysis" (line 155), "However, inferred dated for these" (line 318), "different different scales" (line 542).

Reviewer #3: (No Response)

**Summary and General Comments**

Reviewer #1: This study and the findings presented here provide relevant information for the continued elucidation and understanding of CHIKV and DENV transmission dynamics, and very importantly contribute information from a region of the Americas lacking in genomic representation of these arboviruses.

Reviewer #2: The research manuscript "Convergent trends and spatiotemporal patterns of arboviruses in Mexico and Central

America" by Gutierrez et al. is a sound contribution to the field of arboviral studies that deserves to be published after minor revisions to the text will be made.

Reviewer #3: (No Response)

PLOS authors have the option to publish the peer review history of their article (what does this mean?). If published, this will include your full peer review and any attached files.

Reviewer #1: No

Reviewer #2: No

Reviewer #3: No

Figure Files:

Data Requirements:

Reproducibility:

References

---

## [Decision Letter · Decision Letter 1]

14 Aug 2023

Dear Dr. Escalera-Zamudio,

Thank you very much for submitting your manuscript "Convergent trends and spatiotemporal patterns of Aedes-borne arboviruses in Mexico and Central America" for consideration at PLOS Neglected Tropical Diseases. As with all papers reviewed by the journal, your manuscript was reviewed by members of the editorial board and by several independent reviewers. The reviewers appreciated the attention to an important topic. Based on the reviews, we are likely to accept this manuscript for publication, providing that you modify the manuscript according to the review recommendations. 

Dear Authors,

sincere apologies for the late feedback. 

Thank you for your responses. Please find some minor comments.

Sincerely,

Elvina Viennet, PhD

Section Editor

Dear Authors,

sincere apologies for the late feedback. 

Thank you for your responses. Please find some minor comments.

Reviewer's Responses to Questions

**Key Review Criteria Required for Acceptance?**

**Methods**

-Are the objectives of the study clearly articulated with a clear testable hypothesis stated?

-Is the study design appropriate to address the stated objectives?

-Is the population clearly described and appropriate for the hypothesis being tested?

-Is the sample size sufficient to ensure adequate power to address the hypothesis being tested?

-Were correct statistical analysis used to support conclusions?

-Are there concerns about ethical or regulatory requirements being met?

Reviewer #2: (No Response)

Reviewer #4: Line 90 I see that Zika was considered in the analysis, but I do not see an explanation for why DENV-3 and DENV-4 weren’t.

Line 131 Sequencing was also obtained from public databases. I am assuming that the sequencing method used was like the authors. Can this be clarified?

**Results**

-Does the analysis presented match the analysis plan?

-Are the results clearly and completely presented?

-Are the figures (Tables, Images) of sufficient quality for clarity?

Reviewer #2: (No Response)

Reviewer #4: Line 213 Discussion about the discrepancy between genomic and epidemiological data could move to the discussion section and expanded upon.

Line 226 seems like a repetition. This was already mentioned in line 29-30

Line 319 was there any attempt to control for the sampling biases like with a sensitivity analysis?

Line 432 what are the minority of the serotyped cases? DENV-3 and DENV-4? It would be helpful to state this.

**Conclusions**

-Are the conclusions supported by the data presented?

-Are the limitations of analysis clearly described?

-Do the authors discuss how these data can be helpful to advance our understanding of the topic under study?

-Is public health relevance addressed?

Reviewer #2: (No Response)

Reviewer #4: Line 555 what are these high-risk states?

Line 605 seems to suggest vaccine trials among dengue-naïve individuals. However, vaccines like Dengvaxia are only given to people who have previously had dengue and so trials with other vaccines might not be as straightforward.

**Editorial and Data Presentation Modifications?**

Reviewer #2: (No Response)

Reviewer #4: Minor Revision

**Summary and General Comments**

Reviewer #2: The issues raised in the first round of review have been addressed and I have no further problems with the manuscript. Congratulations to the authors for this great work!

Reviewer #4: Guiterrez et al. investigated whether there were convergent trends of arboviral evolutionary and epidemiological dynamics in Mexico. Their study also adds to a growing literature on the spatiotemporal patterns of DENV in Southern America.

Abstract and Introduction:

Southwest region includes which parts of Mexico and what is the difference between the southern and southwestern states (line 60 and line 349). It would be helpful to reference how the regions were categorized.

Line 17 comes as a surprise. While targeting populations based on age could hold promise, there is no justification in the introduction prior to this statement.

Line 22, I would suggest citing literature confirming the recent introduction of zika and chikungunya. Or you could move this discussion to the next paragraph which addresses the history of chikungunya.

PLOS authors have the option to publish the peer review history of their article (what does this mean?). If published, this will include your full peer review and any attached files.

Reviewer #2: No

Reviewer #4: No

Figure Files:

Data Requirements:

Reproducibility:

References

---

## [Editor Report · Decision Letter 2]

21 Aug 2023

Dear Marina Escalera-Zamudio,,

We are pleased to inform you that your manuscript 'Convergent trends and spatiotemporal patterns of Aedes-borne arboviruses in Mexico and Central America' has been provisionally accepted for publication in PLOS Neglected Tropical Diseases.

Please address the one comment below from Reviewer #4

Best regards,

Aparna Krishnavajhala, Ph.D.

Academic Editor

Elvina Viennet

Section Editor

Reviewer #4:

Line 432 what are the minority of the serotyped cases? DENV-3 and DENV-4? It would be helpful to state this.

---

## [Editor Report · Acceptance letter]

1 Sep 2023

Dear Dr. Escalera-Zamudio,

We are delighted to inform you that your manuscript, "Convergent trends and spatiotemporal patterns of Aedes-borne arboviruses in Mexico and Central America," has been formally accepted for publication in PLOS Neglected Tropical Diseases.

Best regards,

Shaden Kamhawi

co-Editor-in-Chief

Paul Brindley

co-Editor-in-Chief
